# Effects of Chronic Inflammatory Activation of Murine and Human Arterial Endothelial Cells at Normal Lipoprotein and Cholesterol Levels *In Vivo* and *In Vitro*

**DOI:** 10.3390/cells13090773

**Published:** 2024-04-30

**Authors:** Marion Mussbacher, José Basílio, Barbora Belakova, Anita Pirabe, Elisabeth Ableitner, Manuel Campos-Medina, Johannes A. Schmid

**Affiliations:** 1Department of Vascular Biology and Thrombosis Research, Centre for Physiology and Pharmacology, Medical University of Vienna, 1090 Vienna, Austria; marion.mussbacher@uni-graz.at (M.M.); jose.basilio@meduniwien.ac.at (J.B.); barbora.belakova@meduniwien.ac.at (B.B.); anita.pirabe@meduniwien.ac.at (A.P.); manuel.camposmedina@meduniwien.ac.at (M.C.-M.); 2Department of Pharmacology and Toxicology, Institute of Pharmaceutical Sciences, University of Graz, 8010 Graz, Austria; elisabeth.ableitner@uni-graz.at; 3INESC ID, Instituto Superior Técnico, Universidade de Lisboa, 1000-029 Lisboa, Portugal; 4Institute of Pathophysiology and Allergy Research, Medical University of Vienna, 1090 Vienna, Austria

**Keywords:** arteries, endothelial cells, inflammation, NF-kappa B, transcriptomics, pathway analysis

## Abstract

The activation of endothelial cells is crucial for immune defense mechanisms but also plays a role in the development of atherosclerosis. We have previously shown that inflammatory stimulation of endothelial cells on top of elevated lipoprotein/cholesterol levels accelerates atherogenesis. The aim of the current study was to investigate how chronic endothelial inflammation changes the aortic transcriptome of mice at normal lipoprotein levels and to compare this to the inflammatory response of isolated endothelial cells *in vitro*. We applied a mouse model expressing constitutive active IκB kinase 2 (caIKK2)—the key activator of the inflammatory NF-κB pathway—specifically in arterial endothelial cells and analyzed transcriptomic changes in whole aortas, followed by pathway and network analyses. We found an upregulation of cell death and mitochondrial beta-oxidation pathways with a predicted increase in endothelial apoptosis and necrosis and a simultaneous reduction in protein synthesis genes. The highest upregulated gene was ACE2, the SARS-CoV-2 receptor, which is also an important regulator of blood pressure. Analysis of isolated human arterial and venous endothelial cells supported these findings and also revealed a reduction in DNA replication, as well as repair mechanisms, in line with the notion that chronic inflammation contributes to endothelial dysfunction.

## 1. Introduction

Endothelial cells play a pivotal role in immune defense by recruiting leukocytes to sites of infection or inflammation, but they are also fundamental in the development and progression of atherosclerosis, which is initiated by the deposition of lipoproteins and lipids in arteries [1,2]. It is clear that elevated plasma levels of low-density lipoproteins (LDLs) trigger an accumulation of cholesterol-rich lipids in the subendothelial intima layer, in particular at sites of disturbed flow with low or oscillating shear stress [3,4,5]. Lipids accumulating in the vessel wall are subject to oxidative and enzymatic modification, which can cause an inflammatory activation of endothelial cells [6]. On the other hand, unmodified LDL aggregates might also lead to endothelial activation [7], and cholesterol crystals forming in the subendothelial intima might trigger activation of inflammasomes, leading to the release of IL-1β [4,8] and resulting in an upregulation of adhesion molecules in endothelial cells via the transcription factor NF-κB and the recruitment of leukocytes, initiating a vicious circle of inflammation and atherosclerotic lesion formation [9,10]. Monocytes transmigrate through endothelial cells and start phagocytosing the lipid deposits, which triggers the formation of so-called foam cells. More recent studies also demonstrated that paracrine signaling processes lead to a change of the smooth muscle cells from the contractile to a synthetic phenotype and, furthermore, to a transition to macrophage-like cells, which might be less active in phagocytosis than macrophages themselves [11,12]. There are two major mouse models to mimic this pathogenic process: One using a knock-out of the apolipoprotein E (ApoE), resulting mainly in an increase of very low-density lipoprotein (VLDL) and chylomicrons, and a second one with a deficiency of the LDL receptor, triggering an increase in circulatory LDL [13].

Our own studies indicated that additional inflammatory activation of endothelial cells beyond that, which is triggered by high lipoprotein/cholesterol levels on an ApoE-deficient background after feeding a high-fat diet, aggravates and accelerates the atherosclerotic process—and that this is even true when the endothelial activation is initiated at a later stage when atherosclerosis is already developed [14]. Furthermore, we could demonstrate that selective inflammatory stimulation of endothelial cells increases the overall inflammatory profile of the aorta and has a paracrine effect on smooth muscle cells, fostering their cellular transition towards macrophage-like cells. Nevertheless, to the best of our knowledge, how the arterial system would react to a selective inflammatory activation of endothelial cells at normal lipoprotein and cholesterol levels has not been tested so far. To address that, we employed an ApoE wild-type mouse model, which expresses constitutive active IKK2, the main inflammatory activator of the NF-κB pathway, specifically and in an inducible manner in arterial endothelial cells. These mice were kept on a normal diet and, therefore, under normal lipoprotein and cholesterol levels. Transcriptomic changes in the aorta were analyzed by RNA sequencing and pathway analysis, revealing a multitude of effects, including high induction of the SARS-CoV-2 receptor ACE2, which is also an important counterplayer of ACE in blood pressure regulation [15]. Furthermore, we found signs of endothelial dysfunction and cell death, as well as downregulation of protein synthesis pathways. These results were corroborated by studies of isolated primary human endothelial cells, verifying the ACE2 induction and a higher tendency for cell death, as well as a reduction in genes involved in DNA repair.

## 2. Materials and Methods

### 2.1. Mouse Model

Mice with tamoxifen-inducible, arterial endothelial cell-specific expression of constitutive active IκB kinase 2 (caIKK2) were generated by crossing BMX-CreERT2+/− mice (MGI-ID: 5513853, [16]) with mice that express a loxP-flanked STOP cassette upstream of flag-tagged caIKK2 and EGFP (MGI-ID: 3687199, [17]). Tamoxifen (1 mg/20 g mouse) was intraperitoneally injected for five consecutive days to control (BMX-CreERT2−/− caIKK2 fl/fl) and experiment mice (BMX-CreERT2+/− caIKK2 fl/fl). Mice received standard chow and water ad libitum and were kept group-housed on a regular 12 h/12 h dark/light circle. After a recovery period of 10 days, aortas were carefully dissected, cleaned from perivascular adipose tissue, and snap-frozen into liquid nitrogen. Male and female mice were housed and bred at the Medical University Vienna and were used in this study at ~8–12 weeks of age.

### 2.2. RNA Extraction and Sequencing

Frozen aortas were homogenized in a lysis buffer (PeqGOLD total RNA isolation Kit, Thermo Fisher Scientific, Waltham, MA, USA) using a Precellys^®^ homogenizer (VWR^®^, Randnor, PA, USA). Total RNA was isolated according to the manufacturer’s protocol and the RNA concentration was determined with a NanoDrop (VWR^®^, Vienna, Austria). The NEBNext Poly(A) mRNA MagneticIsolation Module and the NEBNext Ultra™ Directional RNA Library Prep Kit for Illumina, following the manufacturer’s protocol (New England Biolabs, Frankfurt am Main, Germany), were utilized by the Core Facility Genomics at the Medical University of Vienna to prepare sequencing libraries. The Bioanalyzer 2100 (Agilent, Santa Clara, CA, USA) was used to perform QC checks on the libraries using a High Sensitivity DNA Kit (Agilent, # 5067-4626, Santa Clara, CA, USA) to ensure proper insert size, while the Qubit dsDNA HS Assay (Fife Technologies Austria, Vienna, Austria) was used for quantification. The libraries were then pooled, with an average length of 330–360 bp, and sequenced on a HiSeq3000/4000 instrument (Illumina, San Diego, CA, USA) using 1 × 50 bp sequencing mode, producing approximately 36 million reads per sample. The quality of the raw reads was assessed using FastQC v0.11.8 [18], while FastQ Screen v0.13.0 [19] was employed to identify any potential contamination or sample mislabeling, including plasmid expression vectors and ribosomal and mitochondrial sequences. The sequences were then aligned to the mouse reference genome build GRCm38.97 using STAR v2.7.3a [20], reporting only reads that map exactly once to the reference genome (-outFilterMultimapNmax 1), with gene counts performed during alignment using the option –quantMode GeneCounts. SAMtools v1.9 [21] was used to index the generated alignment (BAM) files. RSeQC infer_experiment.py v3.0.1 [22] was utilized to determine how RNA-seq sequencing was configured, particularly regarding how reads were stranded for strand-specific RNA-seq data, through comparing the “strandness of reads” with the “strandness of transcripts”. This allowed for the determination of the column from which gene counts should be obtained from STAR output ReadsPerGene.out.tab files. RSeQC read_distribution.py v3.0.1 [22] was used in combination with a BAM/SAM file and reference gene model to calculate how mapped reads were distributed over genome features such as CDS exon, 5′UTR exon, 3′UTR exon, Intron, and Intergenic regions, which could be useful for checking for DNA contamination by evaluating the reads mapped to intergenic regions. QualiMap v.2.2.2-dev [23] was used to evaluate the quality of the alignment. Finally, the results from the different bioinformatic tools were integrated into a single report using MultiQC v1.7 [24]. RNA sequencing data has been deposited in NCBI’s Gene Expression Omnibus database and is accessible through GEO Series accession number GSE230566.

### 2.3. Bioinformatics and Pathway Analysis

In RStudio Server, we performed differential expression analysis using DEseq2 (v1.28.1) with the Wald-test [25] and adjusted *p*-values for false discovery rate (FDR) using the Benjamini–Hochberg (BH) method [26]. Genes were annotated from Ensembl Gene IDs to Entrez Gene IDs by selecting the gene with the highest overall count for each Entrez Gene ID and keeping only those genes coding for proteins. To ensure reliable results, we filtered out genes with less than 10 reads total before normalization. Raw counts were normalized to library size using the estimateSizeFactors function and transformed with the variance stabilizing transformation (vst) function of DESeq2. We used the biomaRt (v2.44.4) [27,28] and org.Mm.eg.db (v3.11.4) [29] packages for annotation. The reported log2 fold changes (logFC) were obtained using the lfcShrink function with the default settings of DESeq2. Our analysis considered genes with an adjusted *p*-value < 0.05 and an absolute logFC cutoff of 1. For Gene Set Enrichment Analysis (GSEA) [30,31], Over Representation Analysis (ORA) [32], and visualization, we used the clusterProfiler (v3.17.1) [33,34]. For GSEA, we ranked genes by the signed −log10 (*p*-value) with the sign from logFC [35]. ORA analysis was performed on genes with an adjusted *p*-value < 0.05. To gain further insights into the biological context, we uploaded the differentially expressed genes (adjusted *p*-value < 0.05) into QIAGEN’s Ingenuity Pathway Analysis software (Spring release 2022, IPA, Ingenuity System Inc., Redwood City, CA, USA). *p*-values are calculated with Fisher’s exact test and adjusted for multiple testing using the BH method. Only those functions and pathways with adjusted *p*-value < 0.05 were considered. Finally, we used the ggplot2 (v3.3.3) [36] package for data visualization. Dataset GSE77962 [37] was reanalyzed with oligo (v1.60.0) [38] using the RMA [39,40,41] function for background subtraction, quantile normalization, and summarization. Group comparison was done with stat_compare_means from the ggpbur (v0.4.0) [42] using the Wilcoxon method.

Gene networks were also analyzed using the NetworkAnalyst platform [43]. For that purpose, differentially regulated genes were uploaded together with their expression data and compared to the protein–protein interactome using the STRING database with a cut-off set to 500 or 700 (as indicated) and only considering interactions with experimental validation. The resulting first-order network was trimmed to the minimum network that is required to connect all uploaded, differentially regulated genes. This network could then be compared with pathways and functions of the Reactome, KEGG, and Gene Ontology databases. The network was then downloaded as a graphml-file, imported into Cytoscape and further analyzed, as well as visualized using the STRING app of Cytoscape.

To decipher the cellular composition of mixed mouse aorta samples, we employed CIBERSORTx [44] with the signature matrix taken from [45]. The deconvolution of mixed aorta samples was performed using the “Impute Cell Fractions” module using the batch correction S-mode in run mode absolute. All input files were in transcripts per million (tpm).

### 2.4. Isolation and Culture of Human Arterial and Venous Endothelial Cells

Human umbilical vein endothelial cells (HUVECs) or human arterial endothelial cells (HAECs) were isolated from umbilical cords as described in [46] by inserting the needle either into the vein or into an artery, followed by collagenase digestion and isolation of the cells. Both types of cells were grown on gelatin (1%, Sigma-Aldrich, Saint Louis, MO, USA) pre-coated cell culture flasks in an M199-based medium containing 20% fetal bovine serum (FBS, Sigma-Aldrich), 0.4% endothelial cell growth supplement with heparin (ECGS/H, PromoCell, Heidelberg, Germany), 2 mM L-glutamine, 0.1% penicillin, 0.1% streptomycin, and 0.25 μg/mL fungizone (all from Lonza, Visp, Switzerland) under standardized conditions (37 °C, 5% CO_2_, 95% humidity). For inflammatory activation, cells were stimulated with TNFα (50 ng/mL) or IL-1β (10 ng/mL, R&D Systems, Minneapolis, MN, USA) for different time periods, as indicated in the figure legends.

### 2.5. Western Blot Analysis of ACE2 Expression

HUVECs were harvested by trypsinization and centrifuged at 300× *g*, then the obtained pellet was washed twice with ice-cold PBS and lysed with RIPA buffer (Sigma-Aldrich) supplemented with a protease inhibitor cocktail (Bimake, Houston, TX, USA). The protein lysates were separated on a 10% polyacrylamide gel and subsequently blotted on PVDF membranes (Carl Roth, Karlsruhe, Germany). Unspecific binding sites were blocked overnight with 5% skimmed milk (AppliChem, Darmstadt, Germany) in PBS-T. After washing, membranes were incubated overnight with ACE2 (1:250, #MA5-32307, Invitrogen, Waltham, MA, USA) and β-actin (1:1000, #2066, Sigma Aldrich) primary antibodies and with HRP-conjugated anti-rabbit secondary antibodies (1:5000, #NA934, Cytiva-Europe GmbH, Vienna, Austria). The blots were soaked in HRP Western Bright Sirius substrate (Advansta Inc., San Jose, CA, USA) and imaged with an Alpha Innotech (San Leandro, CA, USA) Western Blot imager. Quantification of protein bands was performed with ImageJ software (version 1.53c).

### 2.6. Quantitative PCR

RNA expression of ACE2 in cultured endothelial cells was determined by quantitative real-time PCR (qPCR). HUVECs were seeded as triplicates into 6-well plates and treated for the indicated time (0 h, 5 h, 24 h, or 120 h) with 50 ng/mL TNFα or 10 ng/mL IL-1β. After the given incubation period, RNA was isolated using a Peqgold total RNA kit (VWR International, Vienna, Austria) according to the manufacturer’s instructions. Subsequently, 450 ng of total RNA was reverse transcribed using oligo (dT) 18 (Bioline, London, UK), dNTP (Thermo Fisher Scientific), RNAse Inhibitor, and M-MuLV Reverse Transcriptase (both Lucigen, Middleton, WI, USA). Primers for ACE2 (5′-CCC TGC TCA TTT GCT TGG TG-3′ and 5′-AGA ACT TCT CGG CCT CCT TG-3′) and β-actin (5′-AGA AAA TCT GGC ACC ACA CC-3′and 5′-AGA GGC GTA CAG GGA TAG CA-3′) were used together with SsoAdvanced Universal SYBR Green Supermix (Bio-Rad Laboratories, Vienna, Austria) on a CFX Connect Real-Time System (Bio-Rad) real-time PCR machine using the following protocol: denaturation: 95 °C, 5 s; annealing and elongation: 60 °C, 30 s with a total of 50 cycles, followed by melting curve analysis. Relative changes in gene expression were quantified according to Pfaffl [47]. The PCR efficiency was determined from the exponential part of the amplification curve as in [48] and taken into consideration for quantification.

### 2.7. Cell Death Assays

HAECs or HUVECs seeded into 6-well plates were either left untreated or grown under starvation conditions for 16 h (5% FBS instead of 20%, absence of ECGS growth factors). Part of the cells were treated with TNFα (50 ng/mL) or IL-1β (10 ng/mL) for 24 h or 5 d under normal or starvation conditions. Cells were harvested by trypsinization, washed twice with PBS, and incubated with Annexin V-CF Blue and 7-AAD staining solutions using an Apoptosis Detection Kit (ab214663, Abcam, Cambridge, UK) according to the manufacturer’s recommendations. Stained cell suspensions were analyzed with a CytoFLEX S Flow Cytometer (Beckman Coulter, Indianapolis, IN, USA) and analyzed by CytExpert software (version 2.6, Beckman Coulter).

### 2.8. Immunofluorescent Staining of ACE2 Protein Levels in Arterial Endothelial Cells

HAECs were seeded into gelatin-coated 96-well plates and stimulated with 50 ng/mL TNFα or 10 ng/mL IL-1β for 5 h, 24 h, or 5 d. After the incubation, the cells were fixed with 4% PFA (Sigma-Aldrich) for 10 min. Membrane permeabilization was performed with 0.2% Triton-X 100 (Serva, Heidelberg, Germany) and blocking of unspecific binding sites with 3% goat serum (Abcam). The cells were stained with primary rabbit monoclonal antibody against ACE2 (1:100, #MA5-32307, Invitrogen) and goat anti-rabbit Alexa Fluor 647 (1:500, #A21245, Invitrogen) antibody. Nuclear counterstaining was performed by using 1 µg/mL Hoechst 33258 (Cayman Chemical, Ann Arbor, MI, USA). Fluorescent images were taken on an inverted Olympus IX71 microscope (Olympus, Shinjuku, Japan) with a 10× air objective using the 385 nm excitation line with a 447/60 blue emission filter (DAPI/Hoechst) and the 660 nm excitation line with a 692/40 Far red emission filter for ACE2. For quantification, background correction was performed, and ACE2 staining intensity was normalized to the DNA staining intensity.

### 2.9. Statistical Methods

Statistical methods for results obtained with RNA sequencing are described under “Bioinformatics and pathway analysis”. In vitro experiments with cultured HAECs and HUVECs were performed with at least three biological replicates and analyzed with GraphPad Prism 6.07 or version 10.2.1 using ANOVA and Fisher’s LSD test or Student’s *t*-test to compare treated samples with untreated controls.

## 3. Results

### 3.1. Transcriptomic Analysis of Differential Gene Expression in Murine Aortas with Inflammatory Activation of Endothelial Cells

In order to achieve a specific and selective inflammatory activation of endothelial cells without any notable activation of other cell types of the circulation or the vasculature, we applied the BMX-CreERT2 strain, which expresses tamoxifen-inducible Cre recombinase exclusively in arterial endothelial cells [16,49]. This had been crossed with a strain that contains constitutive active IκB kinase (caIKK2)—the main activator of the inflammatory NF-κB pathway—as a transgene downstream of a loxP-flanked stop-cassette (Figure 1A). The BMX-Cre strain has been demonstrated to express Cre recombinase only in endothelial cells of arteries but not veins [49]. Furthermore, analysis of BMX expression in the large Genevestigator database [50] reveals that it is not expressed in smooth muscle cells (Appendix A). The loxP-stop-loxP-caIKK2 strain has been used by several groups to induce chronic, persistent inflammation in a cell-type specific manner and has been demonstrated to not express the transgene at any detectable level in the absence of Cre recombinase activity [17,51,52]. Upon i.p. injection of tamoxifen (on five consecutive days, Figure 1B), arterial endothelial cells express the transgenes caIKK2 and GFP (Appendix A), resulting in the expression of inflammatory target genes of NF-κB, such as E-selectin, VCAM-1, or IL-6 (Appendix A), while the transgene marker is not expressed in leukocytes (Appendix A). The kinase activity of the constitutive active IKK2 transgene has been shown to be slightly lower than the activity that is reached for endogenous IKK2 with TNFα activation (Appendix A). Furthermore, we performed immunohistochemistry on paraffin sections of isolated and fixed mouse aortas to determine the localization of the transgene expression after tamoxifen-mediated induction. This revealed expression in the endothelial layer on the luminal side of the internal elastic lamina (Appendix A). Based on these data, we are confident that our transgene triggers chronic inflammation, specifically in arterial endothelial cells. Nevertheless, it is likely that paracrine signaling is further altering the biological programs of neighboring cells—which was actually the intention of our experimental model.

In an atherosclerosis model (on an ApoE-deficient background and after feeding a high-fat/cholesterol diet), we had found previously that endothelial inflammatory activation resulted in an accelerated and aggravated development of atherosclerosis, which was characterized by a significant inflammatory profile of the aortic transcriptome and a cellular transition of smooth muscle cells towards a non-contractile, synthetic phenotype and further on towards macrophage-like cells [14].

In the current study, we were interested in elucidating the effects of arterial endothelial inflammation on the aorta in the absence of a deregulated lipoprotein and cholesterol balance—thus, on an ApoE wild-type background and under normal diet—and, therefore, in the absence of atherosclerosis. Murine aortas were isolated ten days after the tamoxifen induction protocol, followed by RNA extraction and sequencing as described in the Section 2. Analysis of differentially expressed genes revealed ACE2, the receptor of the SARS-CoV-2 virus and an important blood pressure regulator [53], as the gene with the highest upregulation and the lowest *p*-value, supporting the notion that a chronic inflammatory state might facilitate the entry of the virus into endothelial cells (Figure 1C). Several NF-κB target genes, such as chemokines, were upregulated as well, confirming the inflammatory effect of the transgene. However, the inflammatory profile was less pronounced as compared to arterial endothelial expression of caIKK2 on an ApoE-deficient background and after a high fat/cholesterol diet [14] (Figure 1D). The differences in the transcriptomic changes between endothelial caIKK2 expression on the ApoE-wildtype versus the ApoE-knockout background are specified for the top 25 up- and downregulated genes in Appendix A.

### 3.2. Biological Functions and Regulatory Networks Altered by Inflammatory Activation of Arterial Endothelial Cells

Results of the differential gene expression analysis were used to compute predictions of activated or inhibited pathways and biological functions using the Ingenuity Pathway Analysis platform (Qiagen). This revealed a significant upregulation of cell death pathways, as well as apoptosis and necrosis of endothelial cells (Figure 2A,B), implying that chronic inflammatory activation of the aortic endothelium might result in cellular dysfunction. Furthermore, the tubulation of endothelial cells was predicted to be activated, consistent with the reported role of inflammation in angiogenesis [54]. A complete list of all biological functions and disease associations predicted to be altered is provided in Appendix A. Apart from death receptor and tumor necrosis factor signaling, pathways of mitochondrial beta-oxidation (ketolysis and ketogenesis) were also predicted to be activated.

Pathways that were computed to be downregulated included several processes of the protein synthesis machinery. This was further corroborated by network analysis employing the NetworkAnalyst platform [43]. For the latter, differentially expressed genes (seed genes) were uploaded to the web platform and compared with the STRING database [55] interactome, from which a minimum network linking all seed genes was computed. This network was further analyzed by comparison with the Reactome and KEGG pathways, downloaded as a graphml-file, and imported into Cytoscape [56] for visualization and assessment of functional enrichments using the STRING app. This analysis confirmed that ribosomal genes and pathways of the translational machinery were downregulated (Figure 3). When we checked for cellular profiles in the transcriptome and their change with endothelial caIKK2 expression, we found that endothelial as well as macrophage markers increased, while vascular smooth muscle cell and fibroblast genes decreased after the inflammatory activation of endothelial cells (Appendix A).

In line with the observed significant upregulation of ACE2, the receptor for entry of SARS-CoV-2 into host cells, our pathway analysis implied an upregulation of coronavirus pathogenesis mechanisms. However, coronavirus replication was predicted to be reduced [57,58]. This aspect was investigated in more detail by network analysis, as shown in Appendix A.

### 3.3. Analysis of Gene Set Enrichments and Over-Representation of Biological Processes

Pathway analysis of differentially expressed genes requires defining thresholds for gene expression changes as well as *p*-values to select genes considered to be relevant. However, smaller changes in expression levels for a whole set of genes that are important for a certain pathway might be missed by that procedure if their changes are below the thresholds. This issue can be addressed by gene set enrichment analysis (GSEA) [30,31], which follows a different computational approach. Genes of treated and control samples are ranked according to differential expression metrics, like fold change and/or statistical significance, and this ranked list is compared with molecular signatures of pathways or biological functions. By that, significant changes in given gene sets can be identified, even in the case where the individual genes are not significantly altered. This type of analysis confirmed the crucial downregulation of protein synthesis pathways—but furthermore revealed a reduction in nonsense-mediated decay processes and an upregulation of mitochondrial functions involving the TCA cycle and respiratory electron transport, as indicated by a positive normalized enrichment score (Figure 4A).

Over-representation analysis (ORA) [32] is a widely used method in computational biology to identify enriched biological pathways, gene sets, or other functional categories within a list of genes that are differentially expressed or associated with a phenotype of interest. The method involves the statistical evaluation of the over-representation of the gene set of interest in the reference set, taking into account factors such as the size of the gene set, the total number of genes in the reference set, and the expected frequency of genes in the gene set by chance. By identifying enriched functional categories or pathways, ORA can provide valuable insights into the underlying mechanisms of biological processes. This complementary approach showed the enrichment of downregulated genes in gene sets related to translation and ribosome assembly, as well as N-glycosylation, integrin-signaling, ER organization, and protein localization to the plasma membrane (Figure 4B).

### 3.4. Validation of Results in Primary Human Endothelial Cells from Both Arteries and Veins

Since our transcriptomic analysis of whole aortas from genetically modified mice cannot reveal whether certain genetic alterations occur in endothelial cells expressing the inflammatory transgene or in other cells of the vasculature, we aimed to validate the most important results in isolated primary endothelial cells. Furthermore, we intended to corroborate that the results are also relevant for human cells and not only for murine cells. First, we wanted to test whether ACE2, the highest upregulated gene in the aorta, is also elevated in human endothelial cells. This is highly relevant, as a transcriptional induction of this SARS-CoV-2 receptor in endothelial cells would be expected to facilitate further infection of organs distant from the lungs by spreading the virus via the vasculature. Primary endothelial cells from umbilical cord arteries (human arterial endothelial cells, HAECs) were used as a human analog to the arterial mouse endothelial cells. We investigated potential alterations of ACE2 expression after treatment with TNFα of IL-1β on both the RNA and the protein level using quantitative reverse-transcription PCR and immunostaining of cells, respectively. We found a moderate but significant upregulation of ACE2 mRNA after treatment with IL-1β for either 24 h or 5 d (Figure 5A). TNFα treatment had a minor but insignificant effect on the ACE2-mRNA. However, on the protein level, we observed a significantly elevated ACE2 upon TNFα treatment (Figure 5B). Since we expected that virus uptake would rather occur under low shear stress of veins, we decided to investigate, in addition, venous endothelial cells isolated from human umbilical cords. When we analyzed HUVECs cultured in the presence of TNFα as an inflammatory trigger, we observed an upregulation of ACE2 mRNA only after prolonged treatment as assessed by quantitative PCR (Figure 5A). This could be further verified on the protein level by Western Blot analysis, revealing a pronounced increase in ACE2 protein after 5 days (Appendix A).

Next, we wanted to test whether inflammatory activation of endothelial cells has an effect on cell death (necrosis and apoptosis), as indicated by pathway analysis in murine aortas. Using HAECs, we detected an increased number of apoptotic cells 24 h after treatment with TNFα and 5 d after the addition of IL-1β (Figure 6A). Similar results were found for HUVECs, showing an increase in late apoptotic/necrotic cells treated for 5 days with TNFα as compared to control cells (Figure 6B and Appendix A). Altogether, this supports the pathway analysis performed for the transcriptome of whole aortas, indicating that inflammatory activation of endothelial cells increases their susceptibility to cell death.

### 3.5. Transcriptomic Changes in Human Endothelial Cells after Treatment with TNFα or IL-1β

After verifying that ACE2 is upregulated in human primary endothelial cells following inflammatory activation, we were interested in analyzing the overall change in gene expression under these conditions—and comparing the results with the transcriptomic analysis of whole aortas. To that end, HUVECs were treated both for a short time (5 h) and a long time (9 days) with TNFα (50 ng/mL), followed by extraction of RNA and RNA sequencing (with triplicates for each group). Differentially regulated genes were then analyzed by gene set enrichment analysis, looking for significant alterations in gene ontologies, as well as in KEGG- and Reactome pathways. After both short- and long-term treatment with TNFα, we found the expected upregulation of the inflammatory response and NF-κB target genes (Appendix A). After short-term TNFα treatment, ribosome biogenesis and protein synthesis functions seemed downregulated, similar to the situation in entire aortas. However, this was less prominent after the long-term treatment. DNA replication and repair pathways appeared as downregulated under both inflammatory conditions, which might prime the cells for apoptosis, senescence, or necrosis. However, cell death processes were not among the top-regulated pathways, which may explain why the pro-apoptotic effect of TNFα treatment was rather mild in isolated HUVECs (Figure 6). Comparing the results of isolated HUVECs with those of whole aortas, we assume that cellular crosstalk and paracrine signaling in the vasculature might aggravate the inflammatory response, leading to more pronounced endothelial dysfunction and cell death *in vivo*. In order to investigate the effect of inflammatory activation on isolated human arterial endothelial cells, we analyzed an existing data set describing IL-1β treatment of HAECs [59] in more detail. This revealed an overlap of upregulated genes when compared with the differentially regulated genes in the murine aorta with persistent endothelial cell activation (Figure 7A). Pathway analysis of this dataset again showed the expected prominent upregulation of the inflammatory signaling pathways—but also an increase in apoptosis and cellular senescence, supporting the observed cell death increase in the aorta dataset. In addition, we observed a downregulation of cell cycle and DNA repair mechanisms (Figure 7B).

## 4. Discussion

Chronic inflammatory activation of arterial endothelial cells had marked effects on the transcriptome of murine aortas. However, while we had observed a pronounced inflammatory signature of this genetic stimulation on an ApoE-deficient background after feeding a high fat/cholesterol diet [14], we found a lower inflammation profile on the ApoE-wildtype background on a normal diet, which confirms the additional pro-inflammatory effect of the ApoE deletion [60] as compared to a knock-out of the LDL receptor [13]. Still, it seems clear from our prior studies that inflammation has an aggravating effect for atherosclerosis on top of high lipoprotein levels—an aspect that might contribute to increasing levels of atherosclerosis over the lifespan, as higher age is associated with higher levels of constitutive inflammation—a process often designated as “inflammaging”. The elevated endothelial cell death pathways that we observed in our model, independent of changes in lipoprotein metabolisms, might contribute to plaque erosion and, thus, atherothrombosis.

Interestingly, ACE2, the entry receptor of the SARS-CoV-2 virus, turned out to be the most prominently upregulated gene after the inflammatory activation of arterial endothelial cells. This is in contrast to a prior observation, which claimed that ACE2 is repressed by inflammatory NF-κB activity in renal epithelial cells of rats [61], implying cell-type specific effects of inflammation on the expression of ACE2. The observation of increased ACE2 expression that we made for whole aortas of transgene mice could be confirmed for isolated primary endothelial cells from humans, supporting the notion that endothelial cells at least contribute to the elevated ACE2 expression in the aortas. Our observation of elevated ACE2 levels is in line with the notion that inflammation increases the susceptibility to coronavirus infection and might explain why individuals with chronic inflammation are more prone to severe courses of COVID-19 [62,63].

Since obesity also results in a chronic inflammatory state [64], epidemiological data showing a higher disease risk of these individuals might also be explained by a higher level of ACE2 expression in various tissues, as suggested previously [65]. Re-analysis of transcriptomic data from an obesity study [37] supports the notion that ACE2 is more highly expressed in obese persons and declines with weight loss (Appendix A). We speculate that viral infection of lung epithelial cells may result in lytic death of these cells, followed by infection of endothelial cells of lung capillaries (dependent on the expression of ACE2) and the subsequent release of viral particles from dying endothelial cells into the circulation, where they can spread and infect endothelial cells of other vascular beds all over the organism. Furthermore, endothelial cell death is expected to result in a loss of the endothelial cell layer and exposure of subendothelial collagen, VWF, and tissue factor, leading to the activation of platelets and the priming of the coagulation cascade and resulting in thrombus formation. This is in line with observations of thrombotic complications in the capillaries of the lungs, as well as other organs [66,67]. In our mouse model of chronically inflamed arterial endothelium, pathway analysis actually predicted an activation of the COVID pathogenesis pathway. Nevertheless, the viral replication pathway was predicted to be rather downregulated—mostly due to the reduction in tubulin genes, which are required for viral release from host cells. This opens the possibility that infection of host cells might be supported by the inflammatory state while further virus spreading is reduced—thereby generating a feedback mechanism. Upregulation of ACE2 itself is expected to result in negative feedback based on the fact that it has been reported to exert an anti-inflammatory effect [68,69], which would counteract the ongoing inflammation, while the spike protein of SARS-CoV-2 has been reported to drive inflammation via NF-κB signaling [70]. Furthermore, it has to be considered that ACE2 is an important regulator of blood pressure, acting as a counterplayer to ACE by cleaving angiotensin I to angiotensin 1–9 and Ang II to Ang1–7, which has a vasodilating effect [15,53,71]. Therefore, increased expression of ACE2 triggered by endothelial inflammation would be expected to result in vasodilation. This is in line with the fact that inflammatory processes are normally also accompanied by an increase in blood flow to allow leukocytes easier access to inflamed sites of the organism. In the case of a SARS-CoV-2 infection, this could facilitate the spreading of the virus to other sites of the body. On the other hand, it would counteract high blood pressure, which is a predisposing factor for infection or a severe course of COVID-19 [72]. Since ACE inhibitors can also increase the expression of ACE2, it has been questioned whether these frequent drugs might be harmful in the context of a SARS-CoV-2 infection by upregulating the receptor. However, various clinical observations indicated that the medication is also safe in this case and that the beneficial effects on the overall patient state might be more important than potential negative side effects via increased ACE2 expression [53].

A striking observation of our study is that endothelial inflammation resulted in a significant downregulation of genes involved in protein synthesis. The biological meaning of that is still unclear, but it might be beneficial for a cell in a stressed, inflammatory condition to put a halt on general protein synthesis while coping with the stress situation and repairing a potential damage, which could cause cellular dysfunction.

In summary, we found that chronic inflammatory activation of endothelial cells in murine aortas under normal lipoprotein levels upregulated cell death pathways while it decreased genes of protein synthesis. This was supported by studies in isolated endothelial cells, strengthening the notion that chronic inflammation contributes to endothelial dysfunction that might result in the loss of the endothelium, thereby triggering thrombotic processes.

## 5. Conclusions

Elevated lipoprotein and cholesterol levels are known to have an inflammatory effect, which is an important cofactor in atherosclerosis and related cardiovascular diseases. In our study, we aimed to elucidate the consequences of endothelial inflammation in arteries in vivo using a mouse model under normal LDL cholesterol levels. Transcriptomic analysis of whole aortas, including paracrine effects on other cells of the vasculature and infiltrating leukocytes, revealed the expected upregulation of inflammatory pathways but also a prominent increase in cell death pathways, a downregulation of protein synthesis components, and changes in metabolic pathways. Interestingly, the most upregulated gene was ACE2, the SARS-CoV-2 receptor, which is also an important regulator of blood pressure. The results obtained in the mouse model were further supported by the analysis of human endothelial cells of both arterial and venous origin, which confirmed the upregulation of ACE2, as well as an increase in apoptosis and cellular senescence, while cell cycle and DNA repair pathways appeared to be downregulated.

## Figures and Tables

**Figure 1 cells-13-00773-f001:**
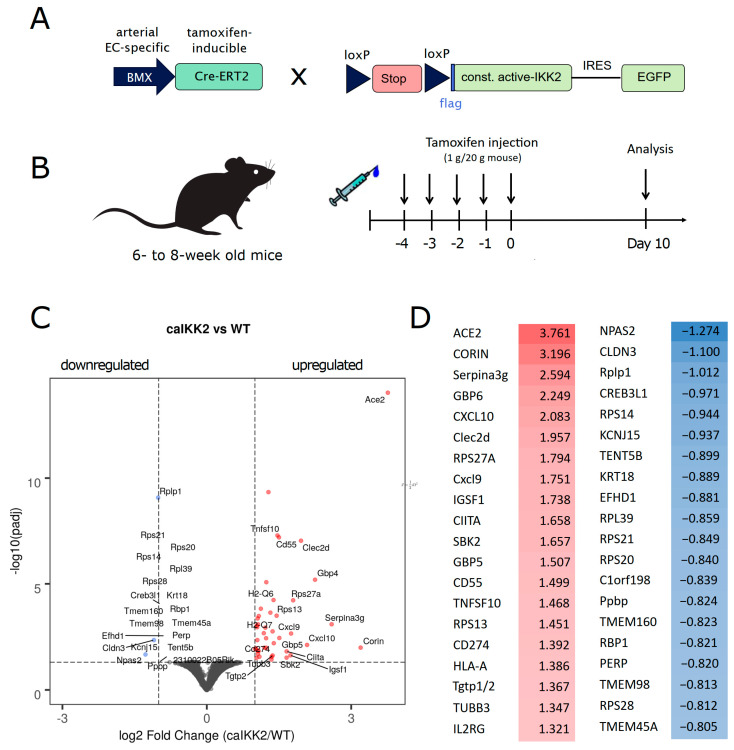
Arterial endothelial inflammation alters the transcriptome of the aorta in a multifactorial manner. (**A**) Scheme of the transgenic mouse model: a mouse strain expressing tamoxifen-inducible Cre recombinase, specifically in arterial endothelial cells, is crossed with a strain holding a constitutive active IKK2 (IKKβ) downstream of a loxP-flanked stop cassette. (**B**) Schematic illustration of inducing chronic inflammation in arterial endothelial cells by the injection of tamoxifen. (**C**) Volcano plot of differentially expressed genes as assessed by RNA sequencing of murine aortas from EC-caIKK2 mice after induction by tamoxifen (as shown in (**B**)) compared to Cre-negative littermate controls (the negative logarithmic value of the adjusted *p*-value is shown on the *y*-axis and the log2-value of the fold change on the *x*-axis; n = 4 for both conditions). (**D**) Top 20 upregulated (red) and downregulated genes (blue).

**Figure 2 cells-13-00773-f002:**
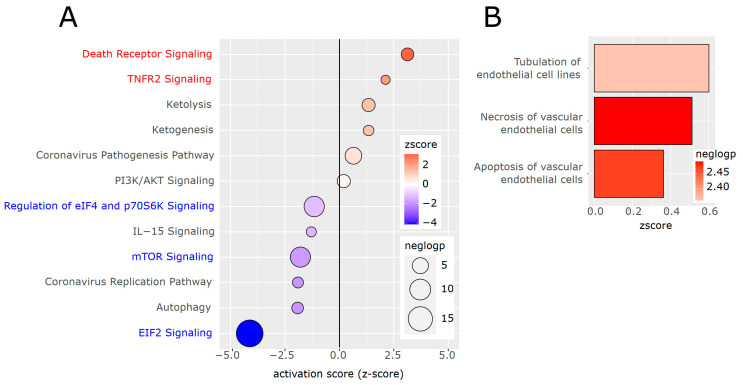
Pathway analysis of differentially expressed genes. (**A**) Differentially regulated genes were subject to Ingenuity Pathway Analysis (IPA, Qiagen). Top canonical pathways, sorted according to the predicted activation score, with the size reflecting the negative log10-*p*-value (neglogp). The colors of the nodes reflect the calculated activation state, the z-score. (Blue text: pathways related to protein synthesis, red text: cell death pathways). (**B**) Biological functions revealed by IPA, filtered for endothelial cell function, and sorted according to the predicted activation score.

**Figure 3 cells-13-00773-f003:**
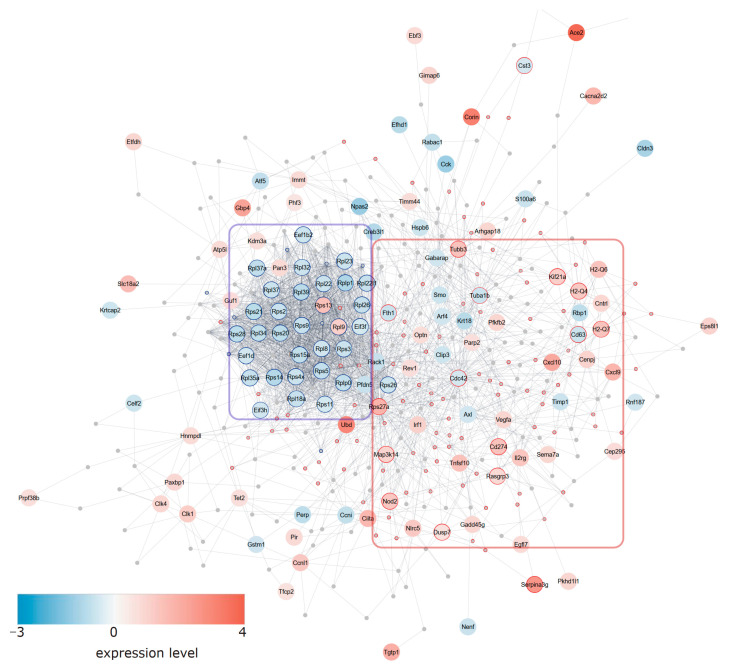
Network analysis of differentially expressed genes. The top 200 differentially expressed genes were subjected to network analyses using the NetworkAnalyst platform. Genes were uploaded with their log2-fold expression values, compared to the STRING interactome (with cut-off 500 and experimental validation), and the minimum network linking all seed genes was computed. The network was saved as a graphml-file and imported to Cytoscape, followed by STRING functional enrichment. Nodes related to protein synthesis are indicated by blue borders and summarized within the blue rectangle. Nodes with a red border and summarized with the red rectangle reflect genes involved in the immune system. The fill color reflects the expression level.

**Figure 4 cells-13-00773-f004:**
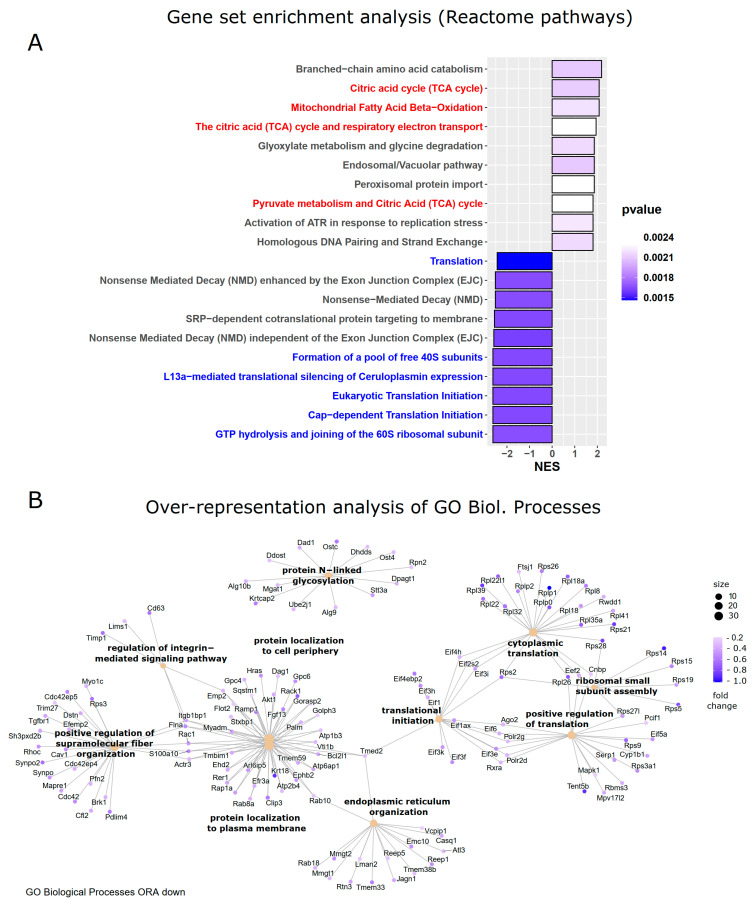
Gene set enrichment and over-representation analysis. (**A**) Genes that are differentially regulated in aortas of mice with inflammatory activation of endothelial cells were subjected to Gene Set Enrichment Analysis (GSEA), which does not require a significant up- or downregulation of individual genes but tests whether certain gene sets related to specific biological functions are significantly altered. The normalized enrichment score (NES) for Reactome pathways revealed a downregulation of translation and protein synthesis gene sets (text in blue) and an upregulation of TCA and mitochondrial functions (in red). (**B**) Over-representation analysis (ORA) of gene ontologies (GOs) for biological processes of downregulated genes confirmed the reduction of protein synthesis pathways.

**Figure 5 cells-13-00773-f005:**
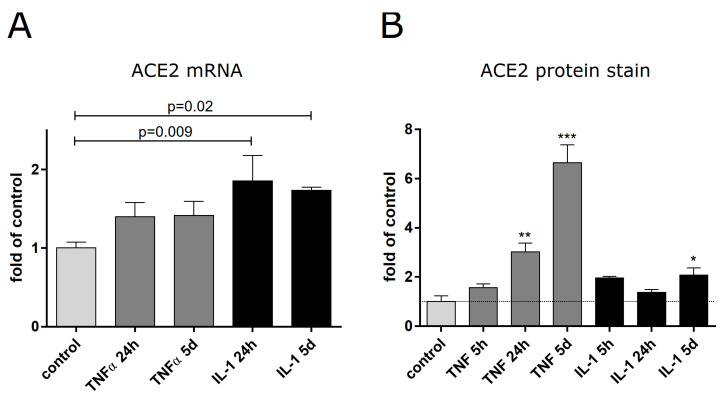
ACE2 expression in human arterial endothelial cells upon inflammatory activation. Human arterial endothelial cells (HAECs) were treated for different periods of time with TNFα (50 ng/mL) or IL-1β (10 ng/mL), followed by RNA extraction. (**A**) RNA was reversed transcribed, and ACE2 mRNA was measured via quantitative PCR (mean ± SEM, n = 3, ANOVA was performed with GraphPad Prism 6.0 with Fisher’s LSD test between control and treated samples; *p*-values as indicated). (**B**) ACE2 protein levels were determined in HAECs treated as indicated by immunofluorescence staining and normalization of the staining intensity to the DNA staining (mean values ± SEM, n = 3; ANOVA and Fisher’s LSD test between treated samples and untreated control; * *p* < 0.05, ** *p* < 0.01, *** *p* < 0.0001).

**Figure 6 cells-13-00773-f006:**
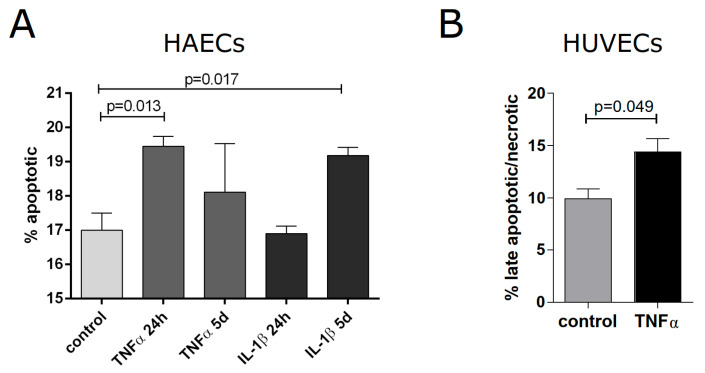
Cell death of human endothelial cells upon inflammatory activation. (**A**) HAECs were treated with TNFα (50 ng/mL) or IL1β (10 ng/mL) for 24 h or 5 d, then trypsinized and stained with a cell death detection kit to assess Annexin V binding; uptake of 7-AAD by permeable cells was also determined using flow cytometry. The percentage of apoptotic (Annexin V-positive) cells was quantified (n = 3, mean ± SEM, Student’s *t*-test for comparisons with the control. Significant differences are indicated by the brackets with the respective *p*-values). (**B**) HUVECs were treated with TNFα (50 ng/mL) for 5 days, followed by quantification of late apoptotic/necrotic cells (positive for Annexin V as well as 7-AAD, Student’s *t*-test, n = 3, mean ± SEM, *p*-value as indicated).

**Figure 7 cells-13-00773-f007:**
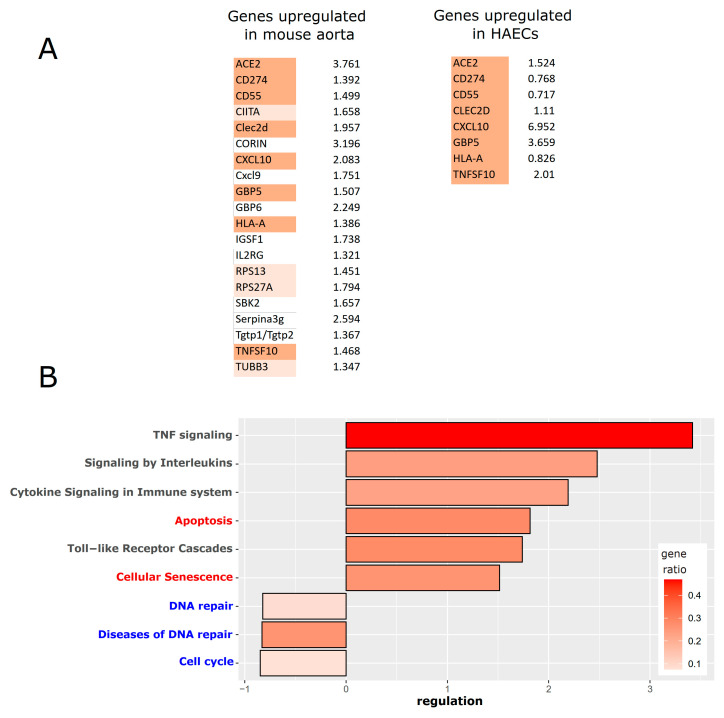
Differentially expressed genes in human arterial endothelial cells (HAECs) after treatment with IL-1β for 4 h (re-analysis of the dataset: GSE89970 from [59]). (**A**) Comparison between the top upregulated genes in murine aorta with persistent inflammatory activation of endothelial cells—and overlapping upregulated genes in cultured human arterial cells after inflammatory activation. Genes that are significantly upregulated in both datasets are highlighted in darker red, and genes of the aorta dataset that are also upregulated in HAECs but do not reach statistical significance are marked in pale red. (**B**) Pathway analysis (based on KEGG and Reactome databases) of differentially regulated genes in HAECs. Inflammatory pathways in black font; apoptosis and senescence pathways in red font are upregulated, while cell cycle and DNA repair pathways (in blue font) are downregulated.

## Data Availability

The datasets generated and analyzed for this study have been uploaded to the Gene Expression Omnibus database and are accessible through GEO Series accession number GSE230566.

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
