# Peer review of "Effects of Chronic Inflammatory Activation of Murine and Human Arterial Endothelial Cells at Normal Lipoprotein and Cholesterol Levels In Vivo and In Vitro"

_cells, 2024, doi:10.3390/cells13090773_

Round 1

Reviewer 1 Report

Comments and Suggestions for Authors

MAJOR

1.     The tittle “Effects of chronic inflammatory activation of endothelial cells in vivo and in vitro2 is ambiguous and it does not reflect the contain of the manuscript. I would suggest to change it, being more specific.

2.     A major concern is that the authors highlight the value of the arterial endothelial cells in their in vivo experiments, however they use HUVEC in the in vitro experiments. Knowing about the great heterogeneity between different endothelial cell types. Validation of the in vitro data should be addressed in HAECs.

3.     About the generation of the conditional knock out mice, some control about the effective activation of IKK2 (IKKβ) in endothelium should be showed in the manuscript.

4.     Authors carry out the RNA array in the whole aortas since the conditional knockout is for endothelium. To really interpretate changes come from the endothelial layer, histological validations should be done to discriminate expression in different cellular types. Validation of your data as protein level are necessary.

5.     Why have authors only focus in ACE2? Other molecules and relative pathways should be molecularly measured in vivo and in vitro.

6.     Why the biology of the system is showed instead the brute date of individual genes from the Transcriptomic study performed in HUVECs?, fig 7.

Comments on the Quality of English Language

The English language is acceptable

Author Response

A point to point reply is provided as attachment. 

Reviewer 2 Report

Comments and Suggestions for Authors

In the submitted manuscript "Effects of chronic inflammatory activation of endothelial cells in vivo and in vitro" Mussbacher et al explore the inflammatory component of the atherosclerotic process. They have used an interesting approach with animal models to explore how inflammation can drive atherosclerosis even if lipid levels are not disturbed. In order to ensure the translational potential of their results, the authors have also tested their hypothesis on human cells. The paper is well-structured and easy to follow despite the complexity of the work done. The methods are well described. However, certain things can be modified to improve the manuscript and its relevance to the CVD audience:
-The title is too general, and more suitable for a review than an original article. I would propose to provide a title that is more specific and includes some of the major findings from the manuscript (for example the major finding is related to ACE2).
-The authors stated in the introduction "It is clear that elevated plasma levels of low-density lipoproteins (LDL) trigger an accumulation of cholesterol-rich lipids in the subendothelial intima layer. This causes an inflammatory activation of endothelial cells, resulting in an upregulation of adhesion molecules via the transcription factor NF-κB and the recruitment of leukocytes, initiating a vicious circle of inflammation [3,4]". This is a little bit of oversimplification and misleading since it overemphasizes endothelial cells, without mentioning other components of inflammatory machinery relevant to atherosclerosis. Please revise.
-The main discovery of the authors revolved around the dysregulation of ACE2 in endothelial cells as a response to inflammation. Although ACE2 is an important piece of the puzzle in hypertension and hypertension is an inseparable part of ASCVD, the authors put complete emphasis on the relevance of ACE2 in SARS-CoV2 infection, not mentioning its possible relevance in hypertension once. This should be revised. 

Patel SK, Velkoska E, Freeman M, Wai B, Lancefield TF, Burrell LM. From gene to protein—experimental and clinical studies of ACE2 in blood pressure control and arterial hypertension. Frontiers in physiology. 2014 Jun 24;5:227.

Bosso M, Thanaraj TA, Abu-Farha M, Alanbaei M, Abubaker J, Al-Mulla F. The two faces of ACE2: the role of ACE2 receptor and its polymorphisms in hypertension and COVID-19. Molecular Therapy-Methods & Clinical Development. 2020 Sep 11;18:321-7.

South AM, Brady TM, Flynn JT. ACE2 (angiotensin-converting enzyme 2), COVID-19, and ACE inhibitor and Ang II (angiotensin II) receptor blocker use during the pandemic: the pediatric perspective. Hypertension. 2020 Jul;76(1):16-22.

-Validation of the results was performed on HUVEC cells. The authors explained, "Since our transcriptomic analysis of whole aortas from genetically modified mice cannot reveal, whether certain genetic alterations occur in endothelial cells expressing the inflammatory transgene or in other cells of the vasculature, we aimed at validating the most important results in isolated primary endothelial cells." and later specify "Since we expected that virus uptake would rather occur under low shear stress of veins, we decided to investigate venous endothelial cells isolated from human umbilical cords". Again, the authors are focusing on the relevance of SARS-CoV2 infection. Since the paper in its current form is not solely dedicated to the relevance of endothelial dysfunction in COVID-19 infection, the focus on veins rather than arteries does not make sense. At least available databases of arteries and or plaques should be checked to see the expression patterns of ACE2. In addition, if possible, the authors should validate ACE2 levels after treatment in endothelial cells of arterial origin. Alternatively, the authors should from the start (title and introduction) give a clear indication that this paper focuses on the relevance of inflammation of endothelial cells in the SARS-CoV2 infection and atherosclerosis. 

Author Response

A point-by-point reply is provided as attachment

Round 2

Reviewer 1 Report

Comments and Suggestions for Authors

Authors have improved the article. However, the reviewer miss a paragraph inside methods section describing the statistical tests used in each part of the manuscript (mainly in cellular approaches). In Figure 5 does not appear number of experiments performed, ..etc..

Comments on the Quality of English Language

I think that the English language is fine.

Author Response

We would like to thank the reviewer for bringing this to our attention. We have now added a paragraph on statistical methods to the Materials and Methods section and also included statistical information into the legends of figures 5 and 6.